# Predictive Whittle Networks for Time Series

**Zhongjie Yu**[*1]      **Fabrizio Ventola**[*1]      **Nils Thoma**[1]

**Devendra Singh Dhami**[1,2]      **Martin Mundt**[1,2]      **Kristian Kersting**[1,2,3]

[1]Department of Computer Science, TU Darmstadt, Darmstadt, Germany
[2]Hessian Center for AI (hessian.AI)
[3]Centre for Cognitive Science, TU Darmstadt

## Abstract

Recent developments have shown that modeling in the spectral domain improves the accuracy in time series forecasting. However, state-of-the-art neural spectral forecasters do not generally yield trustworthy predictions. In particular, they lack the means to gauge predictive likelihoods and provide uncertainty estimates. We propose predictive Whittle networks to bridge this gap, which exploit both the advances of neural forecasting in the spectral domain and leverage tractable likelihoods of probabilistic circuits. For this purpose, we propose a novel Whittle forecasting loss that makes use of these predictive likelihoods to guide the training of the neural forecasting component. We demonstrate how predictive Whittle networks improve real-world forecasting accuracy, while also allowing a transformation back into the time domain, in order to provide the necessary feedback of when the model's prediction may become erratic.

## 1 INTRODUCTION

Time series modeling and forecasting have been a crucial area of research in machine learning, forming a prominent role in its application to several high-impact real-world problems, such as ecological modeling [Recknagel, 2001], finance [Dingli and Fournier, 2017] and healthcare [Alaa and van der Schaar, 2019]. Recent extensions of recurrent neural networks (RNN) [Rumelhart et al., 1985] can achieve impressive performance on complex multivariate time series. However, in many real-world applications, time series are highly subject to several influence factors, which are often hard to capture [Stankevičiūtė et al., 2021]. For example, influence factors could be, or strongly depend on, complex phenomena such as weather conditions or extreme events

like natural calamities or a pandemic. In these cases, the model's forecasts will likely be less accurate. To properly detect such scenarios, a confidence score of the prediction is valuable [Guo et al., 2017] which can make the predictions trustworthy and better support the users in decision-making processes. Whereas several approaches exist, mostly based on Gaussian processes that can also quantify the predictive uncertainty [Seeger, 2004, Rasmussen and Williams, 2006], they are computationally expensive [Bruinsma et al., 2020]. Although one can use hybrid models to scale [Trapp et al., 2020] or add stricter constraints [Corani et al., 2021], these solutions are usually less accurate than current neural counterparts [Alpay et al., 2016]. Recent neural models that operate in the time domain have tackled time series forecasting from a probabilistic perspective, e.g. by making use of neural density estimators [Rasul et al., 2021a] or by employing auto-regressive denoising diffusion models [Rasul et al., 2021b]. These models can be either slow in sample generation or in likelihood computation due to their mostly auto-regressive nature. Moreover, they do not provide a confidence score or a likelihood for a sequence composed of a prediction and its context. Such a measure would enable users to quickly detect potentially problematic forecasts.

Recently, it has been shown that modeling time series in the spectral domain is beneficial in both forecasting accuracy and efficiency since the spectral representation of a time series is generally more compact [Wolter et al., 2020]. Despite being accurate and efficient, current neural spectral time series forecasters do not provide any likelihood score or uncertainty estimate of their predictions in the time domain, nor for an entire sequence like a context with a prediction. On the other hand, although previous probabilistic spectral methods like Tank et al. [2015] and Yu et al. [2021a] have shown improved performance in capturing the distribution of a multivariate time series, they do not tackle forecasting and their predictive power does not outperform established neural architectures. In general, the missing ability to gauge predictive likelihoods is important since it can be exploited during training to learn more accurate forecasters.

---

[*]Equal Contribution

*Accepted for the 38th Conference on Uncertainty in Artificial Intelligence* (UAI 2022).

Motivated by these prior works, we introduce predictive Whittle networks, which integrate a neural spectral forecaster, such as Spectral RNN [Wolter et al., 2020] or a spectral Transformer variant, with Whittle probabilistic circuits (Whittle PCs) [Yu et al., 2021a], i.e. tractable probabilistic models which make use of the Whittle approximation [Whittle, 1953] to facilitate the modeling of the Fourier coefficients of a time series. The aim of predictive Whittle network is to integrate the powerful predictive accuracy of neural spectral forecasters with the useful feedback from tractable and flexible density estimators, in our case, Whittle PCs. We make the following key contributions:

- We propose predictive Whittle networks and the Whittle forecasting loss to exploit the predictive power of spectral neural forecasters and gauge tractable likelihoods from a probabilistic circuit to improve forecasting accuracy.
- We introduce a novel log-likelihood ratio score to provide predictive uncertainty estimates in the time domain based on likelihoods from the spectral domain.

Moreover, to better suit in predictive Whittle networks, we devise improved variants for the neural and probabilistic components.

## 2 RELATED WORK

In a simplified picture, approaches that predict the future course of a time series can be categorized as relying on black-box neural network models, or constructing an elaborate probabilistic model to capture the statistical dependencies among the series' random variables. Intuitively, these perspectives seem to trade-off prediction performance with the ability to accurately gauge data likelihood. We aim to leverage the benefits of both of these views in our work.

**Probabilistic Modeling of Time Series:** A well-known approach to forecasting is to leverage a probabilistic machine learning perspective. For instance, the popular Gaussian processes (GPs) compute probabilistic non-linear regression, allowing exact posterior inference and a natural computation of predictive uncertainty. GPs have been intensively explored for time series regression, classification [Rasmussen and Williams, 2006, Nickisch and Rasmussen, 2008], and have recently been revisited for time series forecasting [Sun et al., 2014, Corani et al., 2021]. Given that GPs do not scale easily, it has been proposed to scale them by employing probabilistic hierarchical mixtures, both for uni-variate [Trapp et al., 2020] and multi-output regression [Yu et al., 2021b].

Alternative generative models which use well-defined likelihood loss functions have thus been proposed. On the one hand, Rangapuram et al. [2018] combined state space models with deep neural networks, while it only centered on forecasting, without modeling the joint distribution of the entire time series. On the other hand, sum-product networks (SPNs) [Poon and Domingos, 2011], a member of the probabilistic circuit (PC) family, have previously been investigated for time series modeling, e.g. dynamic SPNs [Melibari et al., 2016] and the later extension recurrent SPNs [Kalra et al., 2018]. Whereas these approaches now provide tractable and exact probabilistic inference, they have limited representational power because of their strict structural constraints. Thus, they are not as accurate forecasters as deep neural models.

**Neural Spectral Forecasting:** Recurrent neural architectures, such as long short-term memory [Hochreiter and Schmidhuber, 1997] and gated recurrent unit (GRU) [Cho et al., 2014] networks, have paved the way for more accurate neural forecasting. In several scenarios, these approaches have been shown to outperform traditional non-neural models [Siami-Namini et al., 2018]. For instance, N-BEATS [Oreshkin et al., 2019] has achieved great performance on various challenging data sets. Transformers [Vaswani et al., 2017] have been investigated to further improve this forecasting ability in the time domain [Li et al., 2019], with Informer [Zhou et al., 2021] setting the new state-of-the-art at the price of an enormous increase of model size. In a similar spirit, neural auto-regressive models and normalizing flows have been shown to improve predictions [Rasul et al., 2021a, Salinas et al., 2020, Rasul et al., 2021b], but could be difficult to train or slow due to their auto-regressive nature. Recently, spectral RNN [Wolter et al., 2020] has demonstrated that it is beneficial to transform the time series into the spectral domain, in order to obtain a compact and efficient representation that fosters modeling capabilities and yields further performance enhancements. Such spectral modeling has also been pursued in neural sequence prediction with the complex Transformer [Yang et al., 2020]. However, these neural spectral methods do not provide the likelihood of the data, nor of their predictions.

**Probabilistic Spectral Forecasting:** Tank et al. [2015] have introduced a probabilistic approach that works on a spectral representation of stationary time series. They make use of the Whittle approximation to estimate the structure of a graphical model, which encodes the dependencies between the time series components. The Whittle approximation has further been employed in Whittle networks [Yu et al., 2021a], which aim to model the joint distribution of more general non-stationary time series. Whittle networks pose Whittle PCs on top of neural models to inspect their behavior and to capture complex dependencies among the time series components in the spectral domain. They provide the likelihood of an entire time series in the spectral domain but it can not be transformed directly to point-wise likelihoods in the time domain.

In our work, we build on top of the recent advances in all three above lines of research. We propose a hybrid approach that leverages recent insights from modeling in the spectral

domain and combines the benefits of neural forecasters with those of PCs. In this way, we are able to obtain tractable likelihoods and gauge them to further guide training to improve the predictive accuracy.

# 3 PREDICTIVE WHITTLE NETWORKS

In this section, we introduce the predictive Whittle network (PWN). It takes advantage of two distinct elements, namely, a neural spectral forecaster and a Whittle PC, to improve forecasting accuracy and provide useful uncertainty estimates for its predictions. PWN leverages the predictive power of the neural element and gauges the likelihoods from the probabilistic element to weigh its predictions. This is achieved with the Whittle forecasting loss, described in Section 3.1. Then, for each element, we introduce two variants better suited for spectral modeling. Thanks to the flexibility of PWN, they can be used interchangeably. The variants for the neural element are discussed in Section 3.2, while the ones for the probabilistic element are discussed in Section 3.3. A graphical representation of the architecture is shown in Fig. 1. Moreover, in Section 3.4, we present a novel score to provide predictive uncertainties in the time domain. Having such estimates in the time domain is essential to provide intelligible feedback on the predictions.

## 3.1 WHITTLE FORECASTING LOSS & TRAINING

We introduce the Whittle forecasting loss (WFLoss) to gauge likelihood estimates to guide the training of the neural component to superior predictive performance. As represented in Fig. 1, the loss is the connecting element between the neural and the probabilistic components of PWN.

Thanks to its inference capabilities, the Whittle PC can compute the conditional Whittle likelihood $\ell(\mathbf{y} \mid \mathbf{x})$ where $\mathbf{y}$ is a prediction and $\mathbf{x}$ its context. Please refer to Appendix A for further details of Whittle likelihood and Whittle networks. Therefore, to gauge the likelihoods provided for the predictions of the neural forecaster, we propose the WFLoss

$$\text{WFLoss}(\mathbf{x}, \mathbf{y}_{Pred}, \mathbf{y}_{GT}) =$$
$$\frac{1}{M} \sum_{i=0}^{M} (\mathbf{y}_{GT}^i - \mathbf{y}_{Pred}^i)^2 \cdot (\ell_{norm}^{\max} - \ell_{norm}(\mathbf{y}_{Pred}^i \mid \mathbf{x}^i)),$$
$$(1)$$

where

$$\ell_{norm}(\mathbf{y}_{Pred}^i \mid \mathbf{x}^i) = \frac{\ell(\mathbf{y}_{Pred}^i \mid \mathbf{x}^i) - \ell^{\max}}{\frac{1}{M} \sum_{j=1}^{M} (\ell(\mathbf{y}_{Pred}^j \mid \mathbf{x}^j) - \ell^{\max})},$$
$$(2)$$

$\ell^{\max} = \max_k \ell(\mathbf{y}_{Pred}^k \mid \mathbf{x}^k)$, $\mathbf{y}_{Pred}$ denotes the model's prediction while $\mathbf{y}_{GT}$ denotes the ground truth, $\ell_{norm}^{\max} = \max_i \ell_{norm}(\mathbf{y}_{Pred}^i \mid \mathbf{x}^i)$ is the maximum value of the

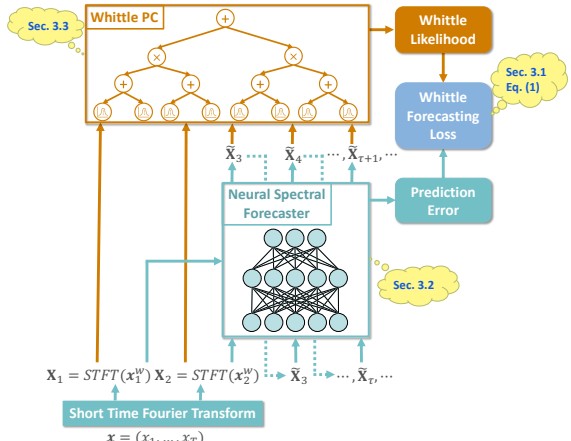

Figure 1: Overview of the predictive Whittle network architecture. The context $\mathbf{x}$ is transformed by STFT with a window $w$, a) flowing as context to the Whittle PC, b) serving as input to the neural spectral forecaster, resulting in the prediction of the Fourier coefficients $\widetilde{\mathbf{X}}_\tau$. These are then provided to the Whittle PC, which uses them with the context to compute the Whittle likelihood. Gauging these likelihood values during training improves forecasting accuracy.

$\ell_{norm}$ in the batch, and

$$\text{MSE}(\mathbf{y}_{GT}, \mathbf{y}_{Pred}) = \frac{1}{M} \sum_{i=0}^{M} (\mathbf{y}_{GT}^i - \mathbf{y}_{Pred}^i)^2 \quad (3)$$

is the mean squared error (MSE). Following Eq. (1), the mean of $\ell_{norm}(\mathbf{y}_{Pred}^i \mid \mathbf{x}^i)$ in a mini-batch equals to 1, hence, the magnitude of the MSE-loss will not be influenced. WFLoss weighs the MSE based on the likelihood computed by the Whittle PC, so that samples with a low likelihood, that are rather in the tails of the distribution, are weighted less than those with a high likelihood. Therefore, our loss formulation prevents the neural spectral forecaster to fit outliers in the data and shifts the focus to data samples that follow the general distribution. However, the likelihood obtained from the Whittle PC is not bounded. Thus, the transformations performed in Eq. (1) and Eq. (2) are necessary for bounding it in $[0, M]$ since it is desirable to weigh the forecasting loss term with bounded weights to avoid numerical issues and improve training stability.

The predictive Whittle network is then trained end-to-end in a coordinate descent fashion. In each optimization step, the weights of the Whittle PC are updated first by maximizing the likelihood of the context and its ground truth prediction, while the neural spectral forecaster's weights remain fixed. Afterwards, the Whittle PC weights are fixed and the neural spectral forecaster is optimized by minimizing the WFLoss. Details and a graphical representation of this alternating procedure can be found in Appendix B.

While training, the Whittle PC may require some epochs until its feedback is valuable for the neural spectral forecaster.

Therefore, we employ a warm-up phase for it, by increasing $\beta \in [0, 1]$ linearly from 0 in the combined loss

$$
\begin{aligned}
\text{Loss}(\beta, \mathbf{x}, \mathbf{y}_{Pred}, \mathbf{y}_{GT}) = {} & (1 - \beta)\, \text{MSE}(\mathbf{y}_{GT}, \mathbf{y}_{Pred}) \\
& + \beta\, \text{WFLoss}(\mathbf{x}, \mathbf{y}_{Pred}, \mathbf{y}_{GT}).
\end{aligned}
\tag{4}
$$

## 3.2 THE NEURAL ELEMENT: SPECTRAL FORECASTER

Here, we present two variants of the neural element which we tailor for spectral forecasting. In Fig. 1, this element is represented as "Neural Spectral Forecaster".

**Spectral RNN (SRNN)** performs recurrent steps over windows retrieved from the short time Fourier transform (STFT) [Wolter et al., 2020]. Details of STFT ($\mathcal{F}_\mathcal{S}$) and its inverse iSTFT ($\mathcal{F}_\mathcal{S}^{-1}$) can be found in Appendix C. Therefore, for a window $\mathbf{x}^w$ with width $T_w$ and a step size $S$, it only has to perform $n_s = (T - T_w)/S + 3$ instead of the typical $T$ time steps for a time series $\mathbf{x} = [x_1, x_2, \cdots x_T]$ of length $T$. The SRNN is defined as follows:

$$
\begin{aligned}
\mathbf{X}_\tau &= \mathcal{F}_\mathcal{S}(\mathbf{x}_\tau^w) \\
\mathbf{z}_\tau &= \mathbf{W}_c \mathbf{h}_{\tau-1} + \mathbf{V}_c \mathbf{X}_\tau + \mathbf{b}_c \\
\mathbf{h}_\tau &= f_a(\mathbf{z}_\tau) \\
\mathbf{y}_\tau &= \mathcal{F}_\mathcal{S}^{-1}(\mathbf{W}_{pc}\mathbf{h}_0, \ldots, \mathbf{W}_{pc}\mathbf{h}_\tau),
\end{aligned}
\tag{5}
$$

with $\tau = [0, n_s]$ enumerating the total number of windows $n_s$. $\mathbf{W}_c, \mathbf{V}_c, \mathbf{b}_c$ and $\mathbf{W}_{pc}$ are weight matrices and $\mathbf{h}_\tau$ is the hidden state. Denote $n_f$ the number of frequencies passing the low-pass filter in STFT. $\mathbf{X}_\tau \in \mathbb{C}^{n_f \times 1}$ is complex-valued, therefore, the RNN cell either needs to operate in the complex space or needs to provide projections $\mathcal{I} : \mathbb{C}^{n_f} \mapsto \mathbb{R}^{n_i}$, $\mathcal{O} : \mathbb{R}^{n_o} \mapsto \mathbb{C}^{n_f}$ for $n_i$-dimensional in- and $n_o$-dimensional outputs respectively.

According to our preliminary experiments (illustrated in Appendix D) and to what has been analyzed in Wolter et al. [2020], operating in the complex space is not substantially beneficial in terms of accuracy for the SRNN. Thus, we employ standard GRU [Chung et al., 2014] with projections. For projections, we use concatenation and splitting respectively, i.e. $\mathcal{I}(\mathbf{X}_\tau) = (Re(\mathbf{X}_\tau), Im(\mathbf{X}_\tau))$ and $\mathcal{O}(\mathbf{h}_\tau) = \mathbf{h}_\tau^{1,\ldots,n_f} + \mathbf{h}_\tau^{n_f+1,\ldots,2 \times n_f} \cdot i$, where $n_i = n_o = 2 \times n_f$. Thus, $\mathbf{h}_\tau \in \mathbb{R}^{n_h \times 1}, \mathbf{W}_c \in \mathbb{R}^{n_h \times n_h}, \mathbf{V}_c \in \mathbb{R}^{n_h \times 2n_f}, \mathbf{b}_c \in \mathbb{R}^{n_h \times 1}$ and $\mathbf{W}_{pc} \in \mathbb{R}^{2n_f \times n_h}$, where $n_h$ is the size of the hidden state. During our preliminary experiments, we have further discovered architectural improvements, i.e. we add residual links [He et al., 2016] to make the network deeper with 2 layers and apply dropout with $p = 0.1$. Details can be found in Appendix D.

**Spectral Transformer (STransformer)** is an architecture tailored for predicting time series in the spectral domain. It is based on the complex Transformer [Yang et al., 2020] which is designed for modeling complex-valued sequences (e.g. Fourier coefficients). However, some of the operations between complex values are only "emulated", e.g. the multi-head attention is emulated with 8 different real-valued attentions between real and imaginary parts.

We propose STransformer as an approach that works natively and holistically on complex numbers. Inputs of the model are the Fourier coefficients given by STFT. We apply positional encoding (PE) per window to preserve the correlation of adjacent frequencies:

$$
\mathbf{X}_\tau = \mathcal{F}_\mathcal{S}(\mathbf{x}_w^\tau) + \text{PE}^\tau,
\tag{6}
$$

where PE is defined as in Vaswani et al. [2017]:

$$
\text{PE}_j^\tau = \begin{cases} \sin(\tau/1000^{2j/d_{\text{model}}}), & \text{if } j \mod 2 = 0 \\ \cos(\tau/1000^{2j/d_{\text{model}}}), & \text{else,} \end{cases}
\tag{7}
$$

where $d_{\text{model}}$ denotes the embedding dimension of the Transformer, and $j$ is the dimension of the positional encoding. To compute $\text{Attention}(Q, K, V) = \text{softmax}^c\left(\frac{QK^T}{\sqrt{d_k}}\right)V$, we shift all computations to the complex space, while employing an alternative softmax in a split-complex fashion [Wolter and Yao, 2018]:

$$
\text{softmax}^c(X) = \text{softmax}(Re(X)) + \text{softmax}(Im(X))i,
\tag{8}
$$

which allows the attention to be distributed over the real and imaginary parts separately. Analogously, we employ complex ReLU [Trabelsi et al., 2018]:

$$
\text{cReLU} = \max(0, Re(X)) + \max(0, Im(X))i.
\tag{9}
$$

For the output $\mathbf{y}_\tau$, we alter the decoding process to allow proper forecasting:

$$
\mathbf{h}_\tau = \mathbf{dec}((\mathbf{X}_\tau, \mathbf{h_0}, ..., \mathbf{h_{\tau-1}})^T, \mathbf{enc}(\mathbf{X}_{0:\tau-1})),
\tag{10}
$$

$$
\mathbf{y}_\tau = \mathcal{F}_\mathcal{S}^{-1}(\mathbf{W_d}\mathbf{h_0}, ..., \mathbf{W_d}\mathbf{h}_\tau).
\tag{11}
$$

The output $\mathbf{y}_\tau$ is computed based on all present and past decoding outputs $\mathbf{h}_\tau$, while **enc** and **dec** denote the encoding and decoding stacks respectively. Thus, now all operations are performed in the complex space. More details on our STransformer and respective preliminary experiments are provided in Appendix E.

## 3.3 THE PROBABILISTIC ELEMENT: WHITTLE PROBABILISTIC CIRCUIT

The Whittle approximation [Whittle, 1953] indicates that the Fourier coefficients of each frequency from a stationary time series are independently complex normal distributed. Recently, Yu et al. [2021a] extended the Whittle approximation to non-stationary time series by introducing the

tractable density estimator called Whittle PCs. We use Whittle PCs as the probabilistic element of the PWN, as depicted in Fig. 1. Here, we consider two variants.

**Conditional Whittle SPN (CWSPN)** has been proposed in Yu et al. [2021a]. To provide a measure of how good a prediction ($\mathbf{y}$) is with respect to a context ($\mathbf{x}$), we aim to model the conditional Whittle likelihood $\ell(\mathbf{y} \mid \mathbf{x})$. Instead of the box window for discrete Fourier transform, in this work, we employ STFT for CWSPNs. Then, the input for the leaves of the CWSPN are the Fourier coefficients of $\mathbf{y}$ in the $\tau^{th}$ window at frequency $k$, i.e., $\mathbf{Y}_\tau^k = \mathcal{F}_S(\mathbf{y})_\tau^k$. To account for the correlations between the real and imaginary parts, they are jointly modeled with a single pairwise Gaussian leaf node, parameterized by a vector of means $\mu_{\mathbf{Y}_\tau^k} \in \mathbb{R}^2$ and a covariance matrix $\Sigma_{\mathbf{Y}_\tau^k} \in \mathbb{R}^{2 \times 2}$. Thus, CWSPN encodes the conditional

$$p(d_1^1, \ldots, d_1^{n_f}, \ldots, d_{n_s}^1, \ldots, d_{n_s}^{n_f} \mid \mathcal{F}_S(\mathbf{x})), \quad (12)$$

where $d_\tau^k = [\mathrm{Re}(\mathbf{Y}_\tau^k), \mathrm{Im}(\mathbf{Y}_\tau^k)]$. Then, based on Eq. (12), we define the conditional Whittle log-likelihood (CWLL) as

$$
\begin{aligned}
\ell(\mathbf{y} &\mid \mathbf{x}) \\
&= \ell(d_1^1, \ldots, d_1^{n_f}, \ldots, d_{n_s}^1, \ldots, d_{n_s}^{n_f} \mid \mathcal{F}_S(\mathbf{x})) \\
&= \log(p(d_1^1, \ldots, d_1^{n_f}, \ldots, d_{n_s}^1, \ldots, d_{n_s}^{n_f} \mid \mathcal{F}_S(\mathbf{x}))),
\end{aligned}
\quad (13)
$$

which models the likelihood of the predicted STFT windows given the STFT windows of the context. The structural constraints of completeness and decomposability of the circuits still hold [Yu et al., 2021a].

**Whittle Einsum Network (WEin)** is our adaptation of Einsum networks [Peharz et al., 2020] for modeling complex values, which is better suited for the spectral domain. We explore Einsum networks since they are a recent efficient implementation of probabilistic circuits. For time series $\mathbf{x}$, WEin models the Fourier coefficients $d_\tau^k$ at frequency $k$ of the $\tau^{th}$ window. Thus, WEin models the joint distribution

$$p(d_1^1, \ldots, d_1^{n_f}, \ldots, d_{n_s}^1, \ldots, d_{n_s}^{n_f}). \quad (14)$$

Therefore, the Whittle log-likelihood (WLL) is defined as:

$$
\begin{aligned}
\ell(\mathbf{x}) &= \ell(\mathcal{F}_S(\mathbf{x})) \\
&= \log(p(d_1^1, \ldots, d_1^{n_f}, \ldots, d_{n_s}^1, \ldots, d_{n_s}^{n_f})),
\end{aligned}
\quad (15)
$$

which models the joint of all STFT windows of a given time series. Given a joint, it is also natural to access the conditional via marginalization: $P_{Y|X}(Y \mid X) = P_{Y,X}(X, Y)/P_X(X)$, where $P_X(X)$ computes the marginal. Thus, although WEin models the joint, given its inference capabilities, it can compute also such conditionals in a tractable way. Therefore, we can employ it in our architecture as Whittle PC in place of the CWSPN (that is learned in a discriminative fashion). In this case, we employ EM for its weight update, since EM is generally more efficient than SGD for such a circuit. More details on our contributions for WEin e.g. multivariate Gaussian leaves are in Appendix F.

## 3.4 PREDICTIVE UNCERTAINTY SCORE

Deep neural models do not naturally provide an uncertainty quantification for their predictions. This is fundamental e.g. for anomaly detection or to identify when the model's predictions might be wrong and, thus, make the predictions more trustworthy. Bayesian methods for neural forecasting, e.g. Liang [2005], have usually focused on model uncertainty and, considering their computational cost, for deeper architectures simpler approximations are necessary [Gal and Ghahramani, 2016]. There are alternative non-Bayesian methods that provide confidence intervals [Stankeviciute et al., 2021] or scores [Brando et al., 2018].

Another way is to take into account the extreme values seen at training time. Here we follow this path and use the notion of likelihood ratios to provide a quantification of the predictive uncertainty. We take advantage of the tractable inference of the Whittle PCs to provide a score that expresses the uncertainty of a prediction by relating its likelihood with the highest training sample likelihood that is used as a reference. Thus, the predictive uncertainty is proportional to the distance between the likelihood of a prediction and the observed maximum likelihood, scaled by the difference between the extreme observed likelihoods.

Crucially, the CWLL already allows estimating the likelihood for a predicted window in the spectral domain. This enables e.g. to take insights into how predictive likelihood changes in the time domain. Thus, we can leverage the window function $w$, to project the likelihood to the time domain at time step $n$:

$$\lambda_{LR}(n) = \max_k \ell(\mathbf{y}^k \mid \mathbf{x}^k) - w(n)\ell(\mathbf{y}_{\mathrm{pred}}(n) \mid \mathbf{x}), \quad (16)$$

where $\ell(\mathbf{y}_{\mathrm{pred}}(n) \mid \mathbf{x})$ denotes the CWLL of a predicted window at time step $n$ given the context, while every other window of the prediction is marginalized, and $\{\mathbf{x}^k, \mathbf{y}^k\}$ is a pair of context and prediction from the training set. To correctly quantify this projection, it is scaled by the distance between the observed maximum and minimum likelihood, defined as

$$\lambda_{LR}^{max} = \max_k \ell(\mathbf{y}^k \mid \mathbf{x}^k) - \min_k \ell(\mathbf{y}^k \mid \mathbf{x}^k). \quad (17)$$

Thus, by using $\lambda_{LR}^{max}$ as a normalization factor for $\lambda_{LR}$, we can estimate the predictive uncertainty with the following log-likelihood ratio score (LLRS):

$$LLRS(n) = \sqrt{|\lambda_{LR}(n)| / \lambda_{LR}^{max}}. \quad (18)$$

In this manner, a likelihood value that is equally low as the worst training sample likelihood (i.e., $\ell(\mathbf{y}_{\mathrm{pred}} \mid \mathbf{x}) = \min_k \ell(\mathbf{y}^k \mid \mathbf{x}^k)$) results in $LLRS = 1$. On training data, larger likelihoods (i.e., $\ell(\mathbf{y}_{\mathrm{pred}} \mid \mathbf{x}) > \min_k \ell(\mathbf{y}^k \mid \mathbf{x}^k)$) result in scores $LLRS < 1$. Therefore, these transformations allow to bound the LLRS of training set samples in $[0, 1]$,

and the LLRS of the test set samples in $[0, +\infty)$ since the worst likelihood could be lower than the worst one observed during training. In practice, given that the CWLL values are unbounded, these transformations make the interpretation and the visualization of LLRS in the time domain easier and more clear. Thanks to the flexible inference of Whittle PCs, we have derived a point-wise uncertainty estimation of the predictions, back in the time domain.

## 4 EXPERIMENTAL EVALUATION

To show the benefits of predictive Whittle networks, we investigate the following research questions.

**(Q1)** Can the uncertainty estimates derived by LLRS be used to distinguish between "good" and "bad" predictions, making the forecasting more trustworthy?

**(Q2)** By gauging predictive likelihood, can predictive Whittle networks improve the forecasting accuracy, outperforming state-of-the-art forecasters?

The experiments have been run on a GPU NVIDIA GeForce GTX 1070 Ti (8GB VRAM) in a system with CPU Intel i7 4x4,0GHz and 32GB RAM. Our code is publicly available.[1]

### 4.1 DATA SETS

We evaluate the model performance on three different real-world data sets and apply z-score normalization to normalize the data for all experiments. The first data set is the *Power* consumption from the European Network of Transmission System Operators for Electricity, with a 15-minute sampling rate, available from Wolter et al. [2020]. The task is to predict 1.5 days of power consumption given 14 days of context. Secondly, we investigate the task of predicting the *Retail* demand, using data from a retail location of a big (national) retailer,[2] spanning over 2 years and including roughly 4000 different products with a daily sampling rate. The task is to predict 6 weeks of products demand given a year of context. Furthermore, we employ the well-known *M4* competition data set [Makridakis et al., 2020]. We use window sizes of 96 on *Power* and 24 on *Retail*. Diverse window sizes are applied on *M4* subsets, which are much smaller, making it more challenging for spectral modeling. The step size of STFT is set to half of the window size for each data set. A more detailed description of the data sets, as well as the window sizes on *M4*, is in Appendix G.

### 4.2 (Q1) USEFUL UNCERTAINTY ESTIMATES

Providing predictive uncertainty in time series forecasting is central. For instance, when performing forecasting in the

long run, the prediction error will likely accumulate, leading the model to produce less accurate forecasts.

The CWLL provided by Whittle PCs can already be used to distinguish between "bad" and "good" predictions. In particular, a lower CWLL indicates a larger MSE ("bad" prediction), since CWLL negatively correlates with MSE, as visualized in Fig. 2. More specifically, to have a quantitative perspective, by selecting the top $5\%$ sequences with the lowest CWLL from the Whittle PC on *Power*, we find that the $75\%$ of all sequences in the top $5\%$ of highest (i.e. worst) MSEs are included. When looking at the top $10\%$ sequences with the lowest CWLL, $98.5\%$ of all sequences that are in the top $5\%$ of highest (i.e. worst) MSEs are included. Therefore, the likelihood by CWLL can inform the user to distinguish between "good" and "bad" predictions.

Considering that CWLL reflects the prediction quality in the spectral domain, we go one step further, by employing LLRS, which can provide predictive uncertainty estimates back in the time domain, and in turn, indicate when the predictions might be erratic or exceptional. To qualitatively evaluate this ability of predictive Whittle networks with LLRS, we run both standard and long-range prediction on both *Power* and *Retail* data sets. For the *Retail* data set, we predict 8 weeks as standard and 32 weeks as long-range prediction, while the model is trained only for 8 weeks prediction. Similarly, for *Power*, we predict 5 days as standard and 40 days as long-range prediction, with the model trained only for 5 days prediction. Fig. 3 depicts the standard prediction together with the predictive uncertainty score estimated with LLRS. For example, on *Retail*, predictive Whittle networks are able to accurately predict the irregular spike around time step 40, providing low uncertainty scores, while it provides higher uncertainty scores from time step 50 on where the prediction slightly differs from the ground truth. Moreover, as shown in Fig. 4 (Left), the LLRS gives relatively lower scores for predictions from time 2000 to 2700 as the prediction matches the ground truth well, and increases considerably after time 3400, as the predictions diverge from the ground truth. On *Retail*, as shown in Fig. 4 (Right), the more the prediction diverges from the ground truth over longer prediction time, the higher LLRS value we obtain, which indicates the increase of predictive uncertainty. Therefore, the LLRS successfully indicates when the prediction is less trustworthy. Note that the LLRS should not be interpreted as a confidence interval or variance, its magnitude reflects the predictive uncertainty measure provided by the PWN. To make this more clear, we provide an alternative visualization of the LLRS in Appendix H.

Both CWLL and LLRS can be computed also on an entire sequence, similar analyses can be done in real-world cases, e.g. to detect if a sequence is likely irregular or to sort sequences w.r.t. their CWLL as a surrogate of their expected error when the ground truth is unavailable. With the feedback for the predictions in the time domain, users can

---

[1]https://github.com/ml-research/PWN

[2]The name of the company cannot be unveiled due to NDA.

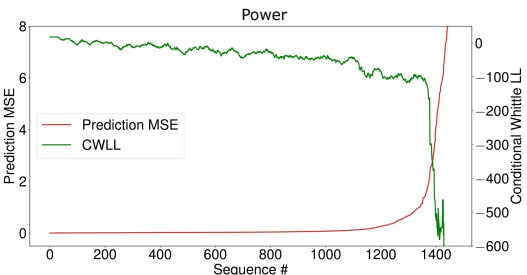
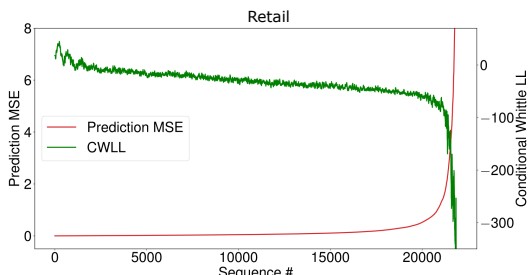

Figure 2: Predictive Whittle networks can correctly separate between "bad" and "good" predictions. This is captured by the correlation of CWLL and MSE on *Power* (Left) and *Retail* (Right). On the x-axis is denoted the enumeration of all test sequences (composed by both context and prediction) in ascending order by MSE. We observe a clear (negative) correlation between a decreasing CWLL and an increasing MSE. The CWLL is smoothed by a moving average of 12 for clarity.

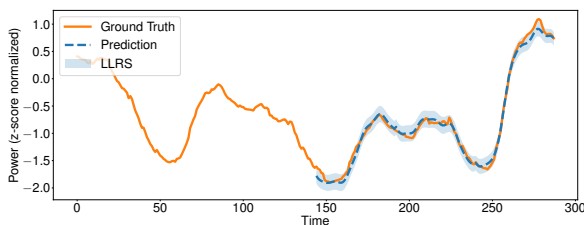
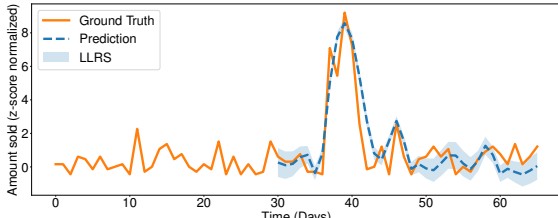

Figure 3: Predictive Whittle networks perform accurate predictions on *Power* (Left) and on a challenging sequence of *Retail* (Right), providing useful predictive uncertainty scores, indicated with LLRS. The context has been cut for clarity.

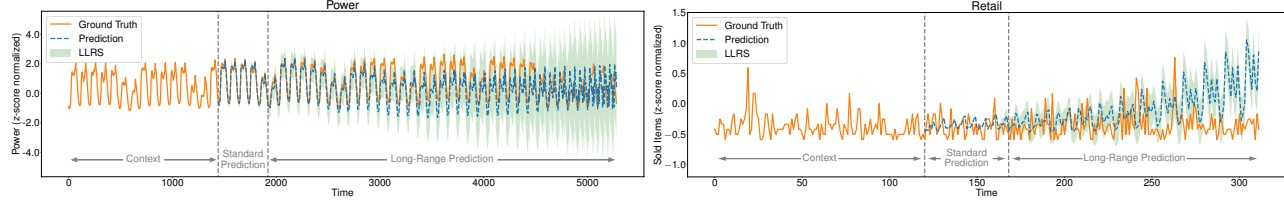

Figure 4: The LLRS from predictive Whittle networks can inform users when the prediction should not be trusted. It increases greatly in the long-range prediction, when prediction is far from the ground truth. The LLRS values at each time step are visualized as bars centered at the corresponding predictions.

have extra knowledge to support decision-making and it is possible to distinguish between potentially "good" and "bad" predictions. We also show additional quantitative analysis by means of a "correlation error" in Appendix I. Therefore, **(Q1)** can be answered affirmatively.

### 4.3 (Q2) ACCURATE FORECASTING

**Setting.** We start by comparing predictive Whittle network to its neural spectral forecasters (SRNN and STransformer). In the same spirit, we compare it also against its Whittle PCs (CWSPN and WEin). For the Whittle PCs, preliminary experiments suggested to project the variance to a fixed interval, i.e. $(10^{-4}, 4)$, which corresponds to a standard deviation interval of $(10^{-2}, 2)$. Note that, for a fair comparison, the neural spectral forecasters have a similar model capacity to predictive Whittle networks, and in turn, have a larger model capacity than PWN's neural components.

Besides the real-world data sets *Power* and *Retail* used to show the ability of predictive Whittle networks to provide useful predictive uncertainty estimates, we test its predictive power also on the challenging *M4* where we use sMAPE as loss term in the WFLoss for training and as the common evaluation metric [Flores, 1986, Makridakis, 1993].

Furthermore, we compare predictive Whittle networks to several neural forecasters. We start with a simple GRU [Chung et al., 2014], operating in the time domain. Then, we compare with DeepAR [Salinas et al., 2020], as a neural probabilistic competitor which also makes use of additional temporal features. Moreover, we compare to N-Beats, another prominent deep neural architecture. It is composed of different blocks specifically designed for time series forecasting [Oreshkin et al., 2019]. Since predictive Whittle networks do not perform model ensembling, for the comparison, we employ the N-Beats singleton model and use a model configuration similar to the default settings.

Table 1: Accuracy in MSE for *Retail*, *Power*, and in sMAPE for *M4*, the lower the better. By both operating in the spectral domain and gauging the likelihoods, predictive Whittle networks outperform strong neural forecasters that operate in the time domain, as visible by the **bold**-face best values. Results include standard deviation across five random-seeded experimental repetitions and runner-up performances are also highlighted in bold if they fall within the best value's range.

| | MSE | | sMAPE on *M4* | | | | |
|---|---|---|---|---|---|---|---|
| | Power kWh $\cdot 10^5$ | Retail Items $\cdot 10^1$ | Yearly (23k) | Quarterly (24k) | Monthly (48k) | Others (5k) | Average (100k) |
| *GRU (Time)* | 23.15 ±1.87 | 3.02 ±0.13 | 15.54 ±0.15 | 11.46 ±0.05 | 13.11 ±0.34 | 4.97 ±0.23 | 12.86 ±0.22 |
| *N-Beats (Time)* | 4.41 ±0.12 | 2.77 ±0.07 | 14.17 ±0.10 | **10.98** ±0.16 | 12.82 ±0.21 | 4.42 ±0.13 | 12.27 ±0.17 |
| *DeepAR (Time)* | 16.83 ±0.60 | 2.74 ±0.02 | 16.88 ±0.33 | 13.26 ±0.51 | 14.83 ±0.32 | 4.85 ±0.10 | 14.43 ±0.36 |
| *Informer (Time)* | **3.77** ±0.09 | 3.03 ±0.04 | 14.49 ±0.18 | 11.96 ±0.43 | 12.97 ±0.22 | 6.33 ±0.97 | 12.75 ±0.30 |
| *CWSPN* | 8.91 ±1.03 | 3.57 ±0.03 | 23.25 ±1.95 | 12.29 ±0.10 | 13.82 ±0.45 | 9.23 ±0.13 | 15.39 ±0.69 |
| *WEin* | 19.28 ±0.61 | 3.72 ±0.05 | 39.34 ±2.28 | 25.91 ±1.30 | 27.07 ±0.48 | 12.20 ±0.48 | 28.87 ±1.09 |
| *SRNN* | 4.16 ±0.06 | 2.43 ±0.06 | 14.25 ±0.06 | 11.23 ±0.06 | **12.59** ±0.04 | 4.77 ±0.06 | 12.26 ±0.05 |
| *STransformer* | 4.14 ±0.08 | 2.70 ±0.04 | 15.22 ±0.51 | 11.24 ±0.22 | **12.56** ±0.14 | 4.67 ±0.02 | 12.46 ±0.24 |
| *PWN (SRNN & CWSPN)* | 4.08 ±0.08 | **2.34** ±0.03 | 14.11 ±0.09 | **10.94** ±0.04 | **12.51** ±0.10 | 4.58 ±0.06 | **12.11** ±0.08 |
| *PWN (SRNN & WEin)* | 3.92 ±0.09 | **2.30** ±0.04 | **14.03** ±0.07 | 11.28 ±0.09 | **12.54** ±0.08 | 4.60 ±0.03 | **12.18** ±0.08 |
| *PWN (STran. & CWSPN)* | 4.01 ±0.08 | 2.66 ±0.05 | 15.19 ±0.27 | **10.92** ±0.16 | **12.56** ±0.09 | **4.49** ±0.06 | 12.37 ±0.15 |
| *PWN (STran. & WEin)* | 3.94 ±0.07 | 2.68 ±0.07 | 15.27 ±0.28 | 11.11 ±0.19 | **12.51** ±0.08 | **4.47** ±0.05 | 12.41 ±0.15 |

To have a fair comparison, we provide all models with a capacity similar to the one of the biggest predictive Whittle network variant. See Appendix J for further details. Finally, we also compare with Informer [Zhou et al., 2021], a state-of-the-art attention-based neural forecaster, with its default settings that result in a model with $11.3M$ parameters, i.e. with a capacity at least 11 times larger than predictive Whittle networks. Given its performance, architecture, and model capacity, we use Informer as a gold standard forecaster. We train each model on *Retail*, *Power*, and *M4* for $9k$, $5k$, $15k$ iterations respectively with a batch size of 256, averaging over 5 random seeds.

There exist widely used metrics for probabilistic forecasting, e.g. CRPS [Matheson and Winkler, 1976, Grimit et al., 2006], MSIS [Gneiting and Raftery, 2007] and quantile loss [Koenker and Bassett Jr, 1978]. These are not applicable in our case as they require the probabilities at each time step of the predictions in the time domain that are not obvious to obtain from the spectral domain. Thus, we evaluate on other two common metrics i.e. MSE and sMAPE.

**Results.** Our results are shown in Table 1. Best results of each data set are marked in bold. We can observe that predictive Whittle networks outperform state-of-the-art models in time series forecasting on all data sets except for *Power* where Informer performs best but employs an 11-times larger model capacity, and predictive Whittle networks achieve competitive performance and outperforms all the other baselines that operate in the time domain. Note that STransformer and SRNN do not use any time series-specific component to account for seasonal changes or similar additional temporal features as e.g., N-Beats or DeepAR, but compared to the baselines they still achieve better or competitive accuracy on almost all the cases. Moreover, predictive Whittle networks can take advantage of its two components and exploits the feedback obtained from the predictive like-lihoods. In this way, it further improves the results of both its Whittle PC and its neural spectral forecaster.

In general, the variants with WEin as Whittle PC form the best setting for predictive Whittle networks that is also the most parameter-efficient one, having remarkably fewer parameters ($\approx 0.6M$) than competitors (ranging from $0.9M$ to $11M$), details are in Appendix J. Regarding WEin, compared to CWSPN, it has additional advantages since it can answer to a broader set of inference tasks and has faster convergence [Peharz et al., 2020]. Arguably, the predictions computed by employing only a Whittle PC, obtained via MPE inference (*CWSPN* and *WEin* in Table 1), are generally not as competitive as the ones obtained with neural forecasters. And given its discriminative nature, in this specific task, CWSPN results more accurate than WEin. For a graphical representation of the results from Whittle PCs, refer to Appendix K.

In summary, our experimental evidence shows that involving a Whittle PC that provides valuable feedback in form of predictive likelihood to predictive Whittle networks can have a significant impact on time series forecasting. We have shown that, in this way, predictive Whittle networks trained with WFLoss improve accuracy over its individual components and also w.r.t. state-of-the-art neural forecasters, thus, answering (**Q2**) affirmatively.

## 5 CONCLUSION

We presented predictive Whittle networks with the Whittle forecasting loss as a method to exploit likelihoods to guide the training process towards more accurate spectral forecasting. They outperform state-of-the-art time series forecasters on challenging data sets. Furthermore, thanks to the novel log-likelihood ratio score we introduced, PWNs also pro-

vide predictive uncertainty estimates in the time domain based on likelihoods from the spectral domain. This is crucial feedback that can signal users and other systems when a prediction is erratic, making the forecasting more trustworthy. Thus, it can foster users in confident decision-making processes in real-world scenarios. This, in turn, can have several implications on multiple scientific fields where time-series forecasting is of paramount importance. For future work, we envision increased involvement of PCs in hybrid deep neural models to push state-of-the-art on challenging tasks. Moreover, since the Fourier transform can be penalized for short window sizes, improving spectral models on such time series is an interesting future direction.

## Acknowledgements

This work was supported by the Federal Ministry of Education and Research (BMBF; project "MADESI", FKZ 01IS18043B, and Competence Center for AI and Labour; "kompAKI", FKZ 02L19C150), the ICT-48 Network of AI Research Excellence Center "TAILOR" (EU Horizon 2020, GA No 952215), the project "safeFBDC - Financial Big Data Cluster" (FKZ: 01MK21002K), funded by the German Federal Ministry for Economics Affairs and Energy as part of the GAIA-x initiative and the Collaboration Lab "AI in Construction" (AICO). It benefited from the Hessian Ministry of Higher Education, Research, Science and the Arts (HMWK; projects "The Third Wave of AI" and "The Adaptive Mind"), and the Hessian research priority programme LOEWE within the project "WhiteBox". The authors thank German Management Consulting GmbH for supporting this work.

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
