# OpenReview forum: "Predictive Whittle Networks for Time Series"
_auai.org/UAI/2022/Conference — UAI 2022 Poster_

### Official Review · Reviewer_ChE2 · 2022-04-10

**Q2(1) Originality/Novelty:** 3
**Q2(2) Significance/Impact:** 3
**Q2(3) Correctness/Technical Quality:** 3
**Q2(6) Clarity Of Writing:** 4
**Q6 Overall Score:** 7
**Q8 Confidence In Your Score:** 2

**Q1 Summary And Contributions:**

The authors use Whittle networks for more accurate spectral forecasting. They outperform state-of-the-art time series forecasters
on challenging data sets.

**Q2 Assessment Of The Paper:**

More detailed information regarding each of these aspects is given below:

**Q2(4) Quality Of Experiments (Optional):**

3: Good: The experimental evaluation is adequate, and the results convincingly support the main claims.

**Q2(5) Reproducibility:**

4: Excellent: Key resources (e.g., proofs, code, data) are available and key details (e.g., proof sketches, experimental setup) are comprehensively described for competent researchers to confidently and easily reproduce the main results.

**Q3 Main Strengths:**

The paper appears to be well written and tackles an important, challenging topic.

**Q4 Main Weakness:**

To fully appreciate several details of the paper, a reader is requested to be familiar with very recent work, notably [Yu et al., 2021a] and, in part, [Peharz et al., 2020].

**Q5 Detailed Comments To The Authors:**

The paper appears to be well written and tackles an important, challenging topic. However, to fully appreciate several details, a reader is requested to be familiar with very recent work, notably [Yu et al., 2021a] and, in part, [Peharz et al., 2020].
I'd recommend the authors consider adding a few additional explanations in the text to help in this respect.

**Q7 Justification For Your Score:**

The paper appears to be technically solid, building on top of recent advancements in probabilistic modelling of time series and probabilistic spectral forecasting, thus with the potential for high impact in that sub-area of AI. The evaluation section appears to be of good quality, the authors linked their code, and I do not envisage significant problems in reproducing the results. The paper does not raise any ethical consideration.

**Q9 Complying With Reviewing Instructions:**

1: Yes.

---

### Official Review · Reviewer_m9F7 · 2022-04-11

**Q2(1) Originality/Novelty:** 2
**Q2(2) Significance/Impact:** 1
**Q2(3) Correctness/Technical Quality:** 3
**Q2(6) Clarity Of Writing:** 2
**Q6 Overall Score:** 3
**Q8 Confidence In Your Score:** 3

**Q1 Summary And Contributions:**

The paper proposes an architecture for probabilistic neural forecasting based on Fourier decomposition and Whittle PCs. Predictive uncertainty is quantified using a likelihood ratio based statistic.

**Q2 Assessment Of The Paper:**

More detailed information regarding each of these aspects is given below:

**Q2(4) Quality Of Experiments (Optional):**

2: Fair: The experimental evaluation is weak: important baselines are missing, or the results do not adequately support the main claims.

**Q2(5) Reproducibility:**

3: Good: Key resources (e.g., proofs, code, data) are available and key details (e.g., proofs, experimental setup) are sufficiently well-described for competent researchers to confidently reproduce the main results.

**Q3 Main Strengths:**

The paper is clearly written. The model, experiments, and training are well described.

**Q4 Main Weakness:**

The paper is lacking background on whittle networks and their relevance to forecasting.

The paper does not relate the proposed likelihood ratio metrics to the usual measures of uncertainty used in probability forecasting.

The evaluation focuses exclusively on the central tendency and not on measures of uncertainty.

**Q5 Detailed Comments To The Authors:**

See Q3&4

**Q7 Justification For Your Score:**

The paper ignores most of the existing probabilistic forecasting literature and does not present experiments quantifying uncertainty prediction.

**Q9 Complying With Reviewing Instructions:**

1: Yes.

---

### Official Review · Reviewer_MAF7 · 2022-04-11

**Q2(1) Originality/Novelty:** 2
**Q2(2) Significance/Impact:** 2
**Q2(3) Correctness/Technical Quality:** 3
**Q2(6) Clarity Of Writing:** 3
**Q6 Overall Score:** 5
**Q8 Confidence In Your Score:** 3

**Q1 Summary And Contributions:**

The authors introduce a frequency-domain time series prediction method that combines a spectral forecaster with a "probabilistic" component based on the Whittle probabilisitic circuit. The two components are jointly trained via a loss function that reweights the standard forecast squared error by a measure of uncertainty based on the probabilistic component. Experiments show competitive prediction performance and reliable indication of when a prediction is likely to be erroneous.

**Q10 Ethical Concerns (Optional):**

None.

**Q2 Assessment Of The Paper:**

More detailed information regarding each of these aspects is given below:

**Q2(4) Quality Of Experiments (Optional):**

3: Good: The experimental evaluation is adequate, and the results convincingly support the main claims.

**Q2(5) Reproducibility:**

3: Good: Key resources (e.g., proofs, code, data) are available and key details (e.g., proofs, experimental setup) are sufficiently well-described for competent researchers to confidently reproduce the main results.

**Q3 Main Strengths:**

- The authors give a clear overview of recent developments in deep methods for frequency-domain modeling and forecasting. This helps frame their contribution as a straightforward approach to combining both forecasting-focused and more general deep probabilistic models for frequency-domain data to both improve forecast quality and equip predictions with some measure of uncertainty.

- The method is thoroughly detailed across both the main paper and appendix sections.

- The experiments are clearly presented in terms of the research question, setup, and evaluation criteria. The authors investigate not only whether their method improves prediction performance, but also whether the probabilistic component can be used to identify when a given prediction is likely to be of low quality.

- The results give evidence for affirmative answers to both research questions. Prediction results are compared against competitive baselines.

**Q4 Main Weakness:**

- The "probabilistic" component of the Whittle prediction network seems to be leveraged in a relatively ad-hoc way - specifically, through a somewhat convoluted procedure to obtain weights that subsequently scale the individual terms in a standard forecasting loss.

- The definition of the Whittle forecasting loss in Eq (1) and discussion of the training procedure in Appendix A raises some questions regarding the training of the proposed WPN model and the interpretation of experimental results in $\S 4.2$.

- The two points above, combined with the prediction results, lead to a seemingly important scientific question that is not ultimately addressed: if incorporating a probabilistic component is so important, why does this method outperform the more principled approach of Whittle PCs?

- The presentation of LLRS results in Figure 4 is poorly described and it is not clear how intervals could have been obtained from these values, or what the interpretation of these intervals might be.


**Q5 Detailed Comments To The Authors:**

More detailed comments and questions related to Q4 above:

- Both the weighting term for the weighted square loss in Eq. (1) and the log-likelihood ratio score ("LLRS") in Eq. (18) are given by somewhat convoluted transformations of the (conditional) log-likelihood. In both cases, it is somewhat unclear why the log-likelihood is being transformed in this way, and if this is important in either a theoretical or practical sense. Some additional discussion by the authors would be valuable here.

- The training procedure (Appendix A) details an alternating procedure where the Whittle PC parameters are first optimized with respect to a log-likelihood criterion, then these are fixed while the forecaster parameters are optimized with respect to the loss function proposed in Eq. (1), and this process is iterated to convergence. But since the Whittle PC parameters don't appear to depend on the NSF parameters, it is not clear why any iteration is required. Why not just fully optimize them first, then optimize the NSF parameters, which depend on them? Perhaps I have simply misunderstood the method, in which case I welcome clarification from the authors.

- Eq. (1) and the associated training procedure raise the question as to how exactly the proposed method has led to improved prediction results. If we first train a model for the likelihood of a given window, and then train a spectral forecaster on a loss weighted by (some function of) the likelihood of these windows, then intuitively it seems that we are focusing the forecaster's training on higher-likelihood windows. Under this interpretation, the method does something akin to soft outlier removal in the spectral domain. Is this a reasonable interpretation, or do the authors suggest a different view? Moreover, doesn't this render the results of $\S 4.2$ somewhat expected, in the sense that we have explicitly trained the forecaster to perform well exactly where the window likelihood is high, and to not worry about errors where the window likelihood is low?

- In a paper with an otherwise high level of presentation quality, Fig 4 stands out as confusing and potentially misleading. It is very much unclear how the "LLRS" quantity, obtained through a somewhat ad-hoc transformation of the window likelihoods, can be converted to an interval value, or what this interval might represent. This is important as the standard interpretation of such time-domain intervals is as a prediction interval under a probabilistic model for a time-domain forecast. It is seriously unclear as to whether or how the LLRS admits such an interpretation, and evidence in the form of proof or simulations is required. I would suggest that the authors add substantial clarification to this part, including the accompanying discussion in $\S 4.2$, or otherwise rework the presentation of the LLRS scores.

**Q7 Justification For Your Score:**

I would welcome some further discussion by the authors on the points made in Q5 above, but overall my impression of the contributions and strengths of this work outweighs that of its weaknesses. My main concern is for the presentation of LLRS results in Figure 4, which I think must be clarified or revised if I am to improve my score.

**Q9 Complying With Reviewing Instructions:**

1: Yes.

---

### Official Review · Reviewer_iCwy · 2022-04-15

**Q2(1) Originality/Novelty:** 2
**Q2(2) Significance/Impact:** 2
**Q2(3) Correctness/Technical Quality:** 3
**Q2(6) Clarity Of Writing:** 4
**Q6 Overall Score:** 6
**Q8 Confidence In Your Score:** 3

**Q1 Summary And Contributions:**

This paper suggests a neural network architecture that both exploits deep learning efficient techniques, while also providing uncertainty estimation through tractable models. The focus is on time series forecasting, where a neural network provides predictive power for forecasting, while a tractable model contributes with a predictive likelihood for uncertainty.
Here, the tractable model acts as a loss function for neural network training.

**Q2 Assessment Of The Paper:**

More detailed information regarding each of these aspects is given below:

**Q2(4) Quality Of Experiments (Optional):**

3: Good: The experimental evaluation is adequate, and the results convincingly support the main claims.

**Q2(5) Reproducibility:**

3: Good: Key resources (e.g., proofs, code, data) are available and key details (e.g., proofs, experimental setup) are sufficiently well-described for competent researchers to confidently reproduce the main results.

**Q3 Main Strengths:**

The use of a probabilistic loss function adds an extra layer of understanding of the model, allowing for a measure of uncertainty in the provided neural network forecasting.

**Q4 Main Weakness:**

The new architecture presented here seems incremental and its possible impact is not very clear from the manuscript.

**Q5 Detailed Comments To The Authors:**

A nice feature of this architecture is it's intrinsic probabilistic semantic. That is, the Whittle PC on top of a neural network provides a sound measure for the neural network uncertainty. As highlighted in Section 3.4, the predictive uncertainty score is “a quantification of the predictive uncertainty.” This is useful information not only during training (as demonstrated by experiments in Section 4) but also serves to support interpretability tools during inference.

I have a curiosity/suggestion point. The Whittle PC seems to be learning the data distribution over a given set of points and its consecutive forecasted points. In practice, it is learning a joint distribution over input and forecast. Thus, one could wonder if learning this joint distribution directly from the dataset would be advantageous. That is, instead of learning over the context (from the dataset) and the forecasted values by the neural network, it would learn only from the given dataset. The idea would be to avoid any neural network bias during an end-to-end training scenario. Any thoughts on this?

Although the ideas in the paper are well executed and the results are encouraging, the contributions here seem incremental. Neither the neural network nor the tractable model component of the architecture is novel. In a way, the architecture itself seems incremental, since computing conditional likelihoods of its input is a well-known inference scenario in tractable models. Thus, the paper would benefit from highlighting challenges specific to assembling this proposed architecture. As well as any theoretical implications/contributions derived from its construction.

Still related to the significance of this work, the manuscript could emphasize how the new architecture's individual components compare (disadvantages and advantages) with compared methods. For example, Figure 2 shows a strong correlation between the conditional Whittle log-likelihood (CWLL) computed in this paper and Mean Squared Error (MSE). While this correlation reassures CWLL as a valid measure, it also diminishes its use - as MSE seems like a similar signal.

**Q7 Justification For Your Score:**

The paper is well-written and provides an architecture that improves time series forecasting while intrinsically measuring the forecast uncertainty. Although experiment results are encouraging, this work could be improved by showing its contribution's relevance and impact.

**Q9 Complying With Reviewing Instructions:**

1: Yes.

---

### Decision · Program_Chairs · 2022-05-15

**Decision:**

Accept (Poster)

**Comment:**

Meta Review: The negative reviewer has major concerns about the insufficient introduction of the whittle network and its evaluation settings. In rebuttal, the authors have addressed them clearly. AC read the paper (not in detail), reviews, and author response, and believes that the paper's strength clearly overweighs its weakness. AC recommends acceptance.